# Beyond the Valve: Incidence, Outcomes, and Modifiable Factors of Acute Kidney Injury in Patients with Infective Endocarditis Undergoing Valve Surgery—A Retrospective, Single-Center Study

**DOI:** 10.3390/jcm13154450

**Published:** 2024-07-29

**Authors:** Christian Dinges, Christiane Dienhart, Katja Gansterer, Niklas Rodemund, Richard Rezar, Johannes Steindl, Raphael Huttegger, Michael Kirnbauer, Jurij M. Kalisnik, Andreas S. Kokoefer, Ozan Demirel, Rainald Seitelberger, Uta C. Hoppe, Elke Boxhammer

**Affiliations:** 1Department of Cardiovascular and Endovascular Surgery, Paracelsus Medical University of Salzburg, 5020 Salzburg, Austria; 2Department of Internal Medicine I, Division of Gastroenterology, Hepathology, Nephrology, Metabolism and Diabetology, Paracelsus Medical University of Salzburg, 5020 Salzburg, Austria; 3Department of Anesthesiology, Perioperative Medicine and General Intensive Care Medicine, Paracelsus Medical University of Salzburg, 5020 Salzburg, Austriam.kirnbauer@salk.at (M.K.);; 4Department of Internal Medicine II, Division of Cardiology, Paracelsus Medical University of Salzburg, 5020 Salzburg, Austriau.hoppe@salk.at (U.C.H.); e.boxhammer@salk.at (E.B.); 5Department of Cardiovascular and Thoracic Surgery, Klinikum Klagenfurt, 9020 Klagenfurt, Austria

**Keywords:** acute kidney injury, infective endocarditis, creatinine, valve surgery

## Abstract

**Background/Objectives**: Infective endocarditis (IE) often requires surgical intervention, with postoperative acute kidney injury (AKI), posing a significant concern. This retrospective study aimed to investigate AKI incidence, its impact on short-term mortality, and identify modifiable factors in patients with IE scheduled for valve surgery. **Methods**: This single-center study enrolled 130 consecutive IE patients from 2013 to 2021 undergoing valve surgery. The creatinine levels were monitored pre- and postoperatively, and AKI was defined by Kidney Disease: Improving Global Outcomes (KDIGO) criteria. Patient demographics, comorbidities, procedural details, and complications were recorded. Primary outcomes included AKI incidence; the relevance of creatinine levels for AKI detection; and the association of AKI with 30-, 60-, and 180-day mortality. Modifiable factors contributing to AKI were explored as secondary outcomes. **Results**: Postoperatively, 35.4% developed AKI. The highest creatinine elevation occurred on the second postoperative day. Best predictive value for AKI was a creatinine level of 1.35 mg/dL on the second day (AUC: 0.901; sensitivity: 0.89, specificity: 0.79). Elevated creatinine levels on the second day were robust predictors for short-term mortality at 30, 60, and 180 days postoperatively (AUC ranging from 0.708 to 0.789). CK-MB levels at 24 h postoperatively and minimum hemoglobin during surgery were identified as independent predictors for AKI in logistic regression. **Conclusions**: This study highlights the crucial role of creatinine levels in predicting short-term mortality in surgical IE patients. A specific threshold (1.35 mg/dL) provides a practical marker for risk stratification, offering insights for refining perioperative strategies and optimizing outcomes in this challenging patient population.

## 1. Introduction

Infective endocarditis (IE), initiated by the invasion of pathogens into the bloodstream, imposes a substantial burden on the cardiovascular system [1]. Its clinical spectrum ranges from subtle to life-threatening scenarios. The implications extend beyond the cardiac realm, with systemic consequences that may necessitate prompt intervention [2,3,4].

Valvular surgery, often required for uncontrolled infection, embolic disease, or severe valvular dysfunction, aims to eradicate the infectious focus, repair or replace valves, and restore cardiac function. Valvular surgery is, therefore, integral to the comprehensive management of IE, aiming not only to address the acute infectious process but also to prevent long-term complications and improve overall patient outcomes [5,6,7].

While valvular surgery stands as a cornerstone in the management of IE, the postoperative period introduces a distinct set of challenges, prominently featuring acute kidney injury (AKI). The kidneys, with their complex vascular network and high metabolic demands, are particularly vulnerable to the hemodynamic shifts and inflammatory responses associated with cardiac surgery [8]. Despite advancements in perioperative care, the occurrence of AKI remains a concerning complication, with the potential to influence both short-term and long-term outcomes.

AKI, defined by a sudden decline in kidney function, poses multifaceted risks to patients undergoing valvular surgery for IE. Beyond its immediate impact on renal function, AKI is associated with increased morbidity and mortality, prolonged hospital stays, and potential long-term renal complications [9]. Understanding the factors contributing to AKI in this specific patient population is crucial for developing targeted interventions that may mitigate its occurrence, ultimately improving postoperative recovery and long-term prognosis.

The decision to delve into the intricate relationships between IE, valvular surgery, and AKI is grounded in the clinical significance of this triad. While valvular surgery is a key component in managing IE, the potential complications, especially AKI, warrant detailed exploration [8,9,10]. AKI, if not managed promptly and effectively, can significantly impact the overall trajectory of patient recovery and may contribute to adverse outcomes.

Moreover, the identification of modifiable factors associated with AKI in IE patients undergoing valve surgery holds promise for informing targeted interventions [11,12]. If specific factors contributing to AKI can be pinpointed and modified, there exists an opportunity to improve renal outcomes and overall survival in this challenging patient group. 

This study, therefore, seeks to unravel the complex dynamics between IE, valvular surgery, and AKI, offering insights that may pave the way for tailored therapeutic approaches and enhanced patient care.

## 2. Materials and Methods

Study Population

In this study, 130 patients from a single, large tertiary center in Salzburg, Austria, who were diagnosed with IE and scheduled for valve surgery, were consecutively enrolled over a nine-year period (2013 to 2021). The focus was exclusively on the incidence of postoperative AKI. Therefore, patients with AKI prior to surgery were excluded from the study. This exclusion criterion encompassed patients with AKI due to potential tubular or glomerular damage or exposure to nephrotoxic antimicrobials. Moreover, patients with kidney failure requiring dialysis prior to surgery were also excluded. To ensure a homogenous study population, all included patients had to exhibit stable renal function with stable renal blood parameters prior to the surgical intervention. Data were analyzed retrospectively. 

Ethics Declaration

The study protocol was approved by the State of Salzburg Ethics Commission (EK: 1109/2023) and conducted in accordance with the principles of the Declaration of Helsinki and Good Clinical Practice. Patient consent was waived due to the retrospective nature of the study, as determined by the State of Salzburg Ethics Commission.

Infective Endocarditis

All patients fulfilled the Duke criteria [13], respectively, criteria stated in the ESC 2015 Guidelines [14] for IE, valid at the time of the indication for surgery.

Indication for Valve Surgery

The indication for cardiac surgical treatment of the valve(s) affected by IE was made by an endocarditis team [13,14,15] consisting of cardiologists, cardiac surgeons, anesthetists and infectiologists. In all cases, one or more of the following conditions was fulfilled: heart failure or severe valve disfunction due to IE, uncontrolled infection, or high risk of embolization.

AKI

The study aimed to assess the incidence of postoperative AKI in an IE population by closely monitoring the creatinine levels before and after surgery. The Kidney Disease: Improving Global Outcomes (KDIGO) criteria for AKI definition were utilized to quantify the occurrence of AKI [16,17]. We used the serum creatinine criteria for the definition and classification of AKI, because urinary excretion data were not continuously and completely available in the medical records of all patients.

Outcomes investigated

Primary Outcomes:Incidence of AKI: The study aimed to determine the frequency of postoperative AKI in IE patients undergoing valve surgery.Relevance of Creatinine Levels for AKI Detection: The investigation sought to establish a practical parameter for risk stratification by assessing the significance of the creatinine levels, particularly focusing on preoperative to the 7th postoperative day.Exploration of modifiable factors contributing to AKI: The study aimed to identify variables amenable to modification that influence the development of AKI. Through a comprehensive assessment of various pre- and intraoperative factors, an attempt was made to uncover potential indicators up to 24 h postoperatively, thus contributing to a more comprehensive understanding of the modifiable elements affecting AKI in patients undergoing valve surgery for IE.

Secondary Outcome:Association of postoperative AKI with short-term mortality: Primary outcomes included evaluating the relationship between AKI development and short-term mortality at specific intervals postoperatively (30, 60, and 180 days).


Statistical Analysis

The statistical analysis and graphical representation were conducted using SPSS (Version 25.0, SPSS Inc., Armonk, NY, USA). To assess the normal distribution of variables, the Kolmogorov–Smirnov–Lilliefors test was employed. Normally distributed metric data were presented as the mean ± standard deviation (SD) and analyzed using an unpaired student’s *t*-test. For not-normally distributed metric data, the median and interquartile range (IQR) were used, and statistical analysis employed the Mann–Whitney *U* test. Categorical data were expressed as frequencies/percentages and compared using the chi-square test. Initially, a Kaplan–Meier curve, accompanied by log-rank tests and numbers at risk, was generated to examine potential differences in 30- to 180-day survival between patients with and without AKI (AKI+ vs. AKI−). To establish creatinine cut-off values at different endpoints (preoperative = D-1, at the day of surgery = D0, and postoperative D1–6), area under the receiver operator characteristics (AUROC) curves were constructed. The analysis included the area under the curve (AUC) and a separate assessment of the Youden Index (YI) based on the presence of AKI. Additional AUROC curves were calculated for predicting various creatinine cut-off values (D-1 to D3) with respect to 30-, 60-, and 180-day mortality. Subsequently, a univariate binary logistic regression model was utilized to calculate the hazard ratio (HR) and 95% CI for factors influencing the presence or absence of AKI. For enhanced comparability, metric data underwent z-transformation. Following this, a multivariable binary logistic regression was conducted to identify independent predictors of AKI. Covariates associated with AKI in the univariate analysis (*p* < 0.050) were entered, and a backward variable elimination was performed. A *p*-value < 0.050 was considered statistically significant throughout the analyses.

## 3. Results

Baseline Characteristics

A tabular overview of the baseline characteristics of the entire cohort, as well as the differentiation between AKI+ and AKI− patients with corresponding significance levels, is provided in Table 1. A total of 130 patients were enrolled in the study protocol. Based on KDIGO criteria, 46 of these (35.4%) exhibited AKI within the first seven post-operative days. The average age of the study population was 61.9 ± 14.4 years, with 70.8% being male. Staphylococci (36.9% of cases) and Streptococci (22.3%) were the most frequently detected germs. In 21 patients (16.2%), no sufficient pathogen evidence was found.

Patients with postoperative AKI were significantly less likely to have Streptococcus-induced IE compared to those with normal renal function (8.7% vs. 29.8%; *p* = 0.006). The total surgery time, clamp time, and perfusion time were significantly prolonged in patients with AKI. The same pattern was observed for ventilation time and length of stay in the ICU. Ultimately, patients with AKI, in addition to a significantly increased preoperative EuroScore II (13.4 ± 9.6 vs. 8.8 ± 10.3; *p* = 0.001), also exhibited significantly higher postoperative 24-h levels of lactate (4.6 ± 2.1 mmol/L vs. 2.4 ± 2.2 mmol/L; *p* = 0.023), Troponin T (1024.0 ± 3619.5 ng/L vs. 728.5 ± 798.3 ng/L; *p* = 0.007), and CK-MB (91.7 ± 117.1 U/L vs. 53.6 ± 28.3 U/L). Additionally, preoperative beta-lactam antibiotics, specifically penicillins, were used significantly more often in patients with AKI (67.4% vs. 45.2%; *p* = 0.016). Similarly, the use of ansamycines was also higher in patients with AKI (26.1% vs. 10.7%; *p* = 0.023).

Creatinine Levels: Second postoperative day as a vulnerable day regarding kidney function

In order to assess the creatinine trajectory both preoperatively and postoperatively, patients underwent laboratory assessment at least once daily. The graphical and numerical representation of the respective creatinine trajectories for the entire cohort, as well as AKI− and AKI+ patients, is depicted in Figure 1 and Table 2.

In the overall population, a baseline creatinine level was observed on the day before surgery (D-1) at 1.1 ± 0.7 mg/dL. Subsequently, postoperatively, the peak was noted on the second postoperative day (D2) with a deviation to 1.6 ± 1.0 mg/dL, followed by a decline to an average of 1.2–1.3 mg/dL in the subsequent course. Patients who did not meet the criteria for AKI in the postoperative period started with an average creatinine of 1.1 ± 0.6 mg/dL, significantly lower than that of AKI patients (1.2 ± 0.7 mg/dL; *p* = 0.003). Furthermore, AKI+ patients maintained significantly higher postoperative creatinine levels compared to AKI− patients throughout the course. The peak was reached on D2 for both groups (1.3 ± 0.7 mg/dL vs. 2.2 ± 1.0 mg/dL; *p* < 0.001).

Kaplan–Meier: AKI as a driving force for premature mortality

Not only were the in-hospital deaths significantly different between the AKI+ and AKI− cohorts (Table 1: 32.6% vs. 7.1%), but the influence of AKI as a driving force for premature mortality also manifested in short-term outcomes over a 180-day period. This observation is evident in the corresponding Kaplan–Meier curve, supported by log-rank tests and the numbers at risk, depicted in Figure 2. Patients experiencing postoperative AKI consistently demonstrated a significantly elevated mortality throughout the follow-up period from 30 to 180 days. The 30-day mortality for AKI+ patients was already at 21.7%, rising to 39.1% after 6 months.

AUROC-AKI: Creatinine of 1.35 mg/dL as a relevant predictor for postoperative AKI

AUROC analyses were calculated to assess creatinine cut-off values on D-1 to D4 in relation to the presence of postoperative AKI (Figure 3).

The most accurate diagnostic prediction for the occurrence of AKI was achieved for the second postoperative day (D2) using a creatinine cut-off of 1.35 mg/dL (AUC: 0.901; 95% CI: 0.849–0.953; sensitivity: 0.89; specificity: 0.79; YI: 0.68; *p* < 0.001). The same cut-off value of 1.35 mg/dL was observed on D0, D1, and D3, with AUC values ranging from 0.781 to 0.886 and *p*-values < 0.001.

AUROC-Mortality: Creatinine of 1.35 mg/dL as a relevant predictor for postoperative mortality

Further AUROC analyses were figured out to analyze the creatinine cut-off values on D-1 to D3 in relation to postoperative mortality (Figure 4: 30-day mortality; Figure 5: 60-day mortality; Figure 6: 180-day mortality).

Once again, the most reliable diagnostic performance was observed on the second postoperative day, with a creatinine cut-off of 1.35 mg and AUC values ranging from 0.708 to 0.789. The corresponding sensitivities ranged from 0.79 to 0.93, specificities from 0.61 to 0.64, YIs from 0.42 to 0.54, and *p*-values were ≤ 0.001.

Binary Logistic Regression: Hemoglobin, CK-MB, and renal excretion 2–3 h after surgery as independent predictors for postoperative AKI

In order to investigate the influencing factors concerning postoperative AKI, a univariate and multivariable binary logistic regression was figured out (Table 3). 

For a clearer overview, only parameters with a *p*-value ≤ 0.050 were listed in the univariate analyses and subsequently included in the multivariable analysis. Ultimately, renal excretion 2–3 h postoperatively, the minimally measured intraoperative hemoglobin in the blood gas analysis, and the maximum CK-MB 24 h postoperatively emerged as independent markers for the occurrence of postoperative AKI.

## 4. Discussion

The comprehensive analysis presented in this manuscript sheds light on critical aspects surrounding AKI in IE patients undergoing valve surgery. Our findings, as outlined in the Section 3, highlight several key observations that warrant thoughtful consideration and further discussion.

Postoperative frequency of AKI in patients undergoing valve surgery due to IE

AKI occurs in approximately 20–30% of heart surgery cases [18,19,20]. The frequency of postoperative AKI in heart surgery varies, but it is particularly common in procedures involving cardiopulmonary bypass or valve surgeries [21]. The intricate nature of these surgeries, coupled with factors like prolonged exposure to anesthesia, inflammation, and ischemia–reperfusion injury due to reduced renal blood flow during surgery, contributes to the increased susceptibility of the kidneys to injury [22,23]. 

In the context of IE, the risk of postoperative AKI may be heightened due to the infectious nature of the condition, potentially exacerbating the inflammatory response and contributing to renal complications [24]. The frequency of postoperative AKI in patients with IE can vary depending on various factors, including the severity of the infection, the extent of cardiac involvement, and individual patient characteristics. While precise figures may vary across studies, research suggests that the incidence of AKI in patients undergoing valve surgery for IE can range from 20% to 40% [13,24,25], with some studies reporting higher rates in more severe cases [9]. The incidence of postoperative AKI in our study population was substantial, with 35.4% of patients experiencing this complication. This underscores the vulnerability of patients with IE to renal complications following valve surgery. In addition, the occurrence of AKI causes a significant increase in health care costs [10]. 

Early detection of AKI: Overcoming the challenge of delay

Given the complexities of cardiac surgical procedures and the vulnerability, especially of patients with IE, to perioperative kidney injury, there is a pressing need for enhanced surveillance strategies during this critical period. Implementing targeted monitoring protocols can help clinicians identify AKI promptly and intervene before irreversible damage occurs. Addressing the challenge of delayed AKI detection in cardiac surgery patients is essential for improving clinical outcomes and mitigating the impact of kidney injury on postoperative recovery [26]. 

Our study highlights the critical importance of overcoming this delay in AKI detection. We identified the peak in creatinine elevation on the second postoperative day as a significant predictor for AKI, with a specific threshold of 1.35 mg/dL demonstrating the optimal diagnostic accuracy. This finding underscores the pivotal role of early postoperative monitoring, particularly on the second day, in identifying patients at risk of AKI and facilitating timely interventions. 

However, the accuracy of traditional biomarkers, such as serum creatinine, can be compromised in the perioperative period, leading to a delayed diagnosis and potential underestimation of the AKI severity [27]. During the immediate postoperative phase, patients often undergo aggressive fluid resuscitation, which can result in dilutional effects on serum creatinine levels. This dilution may mask the true extent of kidney injury, making it challenging to identify AKI promptly [28]. Consequently, there is a risk of missing the optimal window for intervention, potentially exacerbating kidney damage and prolonging recovery time. Given these challenges, extensive nephrological diagnostics, including cystatin C with clearance determination, fractional sodium excretion, and fractional urea excretion, both pre- and postoperatively, are advisable. The kidney may already be compromised by factors such as nephrotoxic antibiotics or endocarditis itself, leading to glomerular or tubular damage, even if renal function has been maintained up to this point. 

This fact reinforces the need for proactive measures during this critical timeframe to effectively manage and improve outcomes for patients at risk of postoperative AKI and, if necessary, to establish other biomarkers for early detection in the first two postoperative days [29].

Consistency of our creatinine cut-off value highlights AKI’s significance in short-term mortality prediction

Our study explored the association between creatinine, AKI, and short-term mortality in IE patients. The examination of creatinine cut-off values in relation to postoperative mortality provides relevant insights. The consistency of the again 1.35 mg/dL cut-off value across multiple time points (D0 to D3) and its association with short-term mortality at 30, 60, and 180 days postoperatively underscore its robust predictive value. 

Postcardiac surgery serum creatinine has been described previously, in a study involving over 6000 patients, by Hou et al. [30] as a robust and versatile outcome indicator. Furthermore, our data support what has been reported by Ye et al. [31], who also found that creatine levels on postoperative day 2 (48 h after cardiac surgery) provided the best prediction of future mortality.

Despite the pitfalls discussed above regarding creatinine determination, these observations underscore the utility of creatinine levels, particularly on the second postoperative day, not only in the detection of AKI but also in the prognosis of short-term mortality.

How can we prevent postoperative AKI following surgery for IE?

In the pursuit of understanding the modifiable factors contributing to AKI, our analyses identified renal excretion 2–3 h postoperatively, the minimum intraoperative hemoglobin, and the maximum CK-MB 24 h postoperatively as independent markers for postoperative AKI. Notably, the significant increase in CK-MB may potentiate the process of AKI rather than merely serving as a marker predicting it. This suggests that the rise in CK-MB could contribute to acute tubular necrosis and exacerbate kidney injury.

Early identification and monitoring of modifiable risk factors, such as renal excretion and intraoperative hemoglobin levels, are crucial for implementing targeted interventions to prevent AKI. Renal function should be assessed promptly postoperatively, with close monitoring of urine output, serum creatinine levels, and renal excretion rates in the hours immediately following surgery. Any decline in renal function or signs of impaired renal excretion should prompt early intervention to optimize hemodynamic stability and renal perfusion [32].

Intra- and postoperatively, strategies to minimize hemodynamic instability and blood loss are essential to preserve renal function. This includes meticulous attention to fluid balance, optimizing the preload, and maintaining adequate perfusion pressure to ensure optimal renal blood flow [33,34,35]. Hemoglobin levels should be closely monitored throughout the surgical procedure, with prompt intervention to address any significant declines to mitigate the risk of renal hypoperfusion and subsequent injury [36].

Avoidance of nephrotoxic agents, such as certain medications and contrast agents, is essential to prevent additional stress on the kidneys. Close collaboration between multidisciplinary teams, including anesthesiologists, nephrologists, cardiologists, and cardiac surgeons, is vital for optimizing perioperative care and minimizing the risk of AKI. Furthermore, the integration of artificial intelligence (AI), as demonstrated in the study by Kalisnik et al. [37], could revolutionize early AKI detection, thereby enhancing perioperative care outcomes in patients undergoing cardiac surgery.

## 5. Limitations

While our study provides valuable insights into the association between postoperative AKI and IE patients undergoing valve surgery, several limitations should be acknowledged. The study’s retrospective design introduces potential biases and relies on existing records, limiting its control and data collection. Being a single center, its findings may lack broader applicability to diverse patient populations. Additionally, the small sample size of 130 patients could compromise the statistical power and generalizability. Especially, an evaluation regarding the surgical indication (heart failure, uncontrolled infection, or high risk of embolization) could not be made from the retrospective data. Conclusions from a further subgroup analysis would probably not reach statistical significance due to the relatively small number of general cases. The focus on valve surgery for IE may not extend to other surgical procedures with different AKI risk profiles. Importantly, various etiologies of AKI in the context of endocarditis, such as glomerular damage, tubular damage, the use of nephrotoxic medications (especially antibiotics), and the impact of septic emboli, were not diagnostically evaluated. This underscores the necessity for prospective studies to thoroughly investigate these factors. Lastly, while the KDIGO criteria for AKI are widely accepted, they may not capture the full spectrum of renal dysfunction. Lastly, while the KDIGO criteria for AKI are widely accepted, they may not capture the full spectrum of renal dysfunction.

## 6. Conclusions

This retrospective study of patients with IE undergoing valve surgery highlights the substantial incidence of postoperative AKI. The identification of a specific creatinine threshold (1.35 mg/dL) on the second postoperative day as a robust predictor for both AKI and short-term mortality offers a practical parameter for risk stratification. These findings emphasize the importance of vigilant monitoring, early intervention, and the potential for targeted interventions to enhance renal outcomes and overall survival in this challenging patient population.

## Figures and Tables

**Figure 1 jcm-13-04450-f001:**
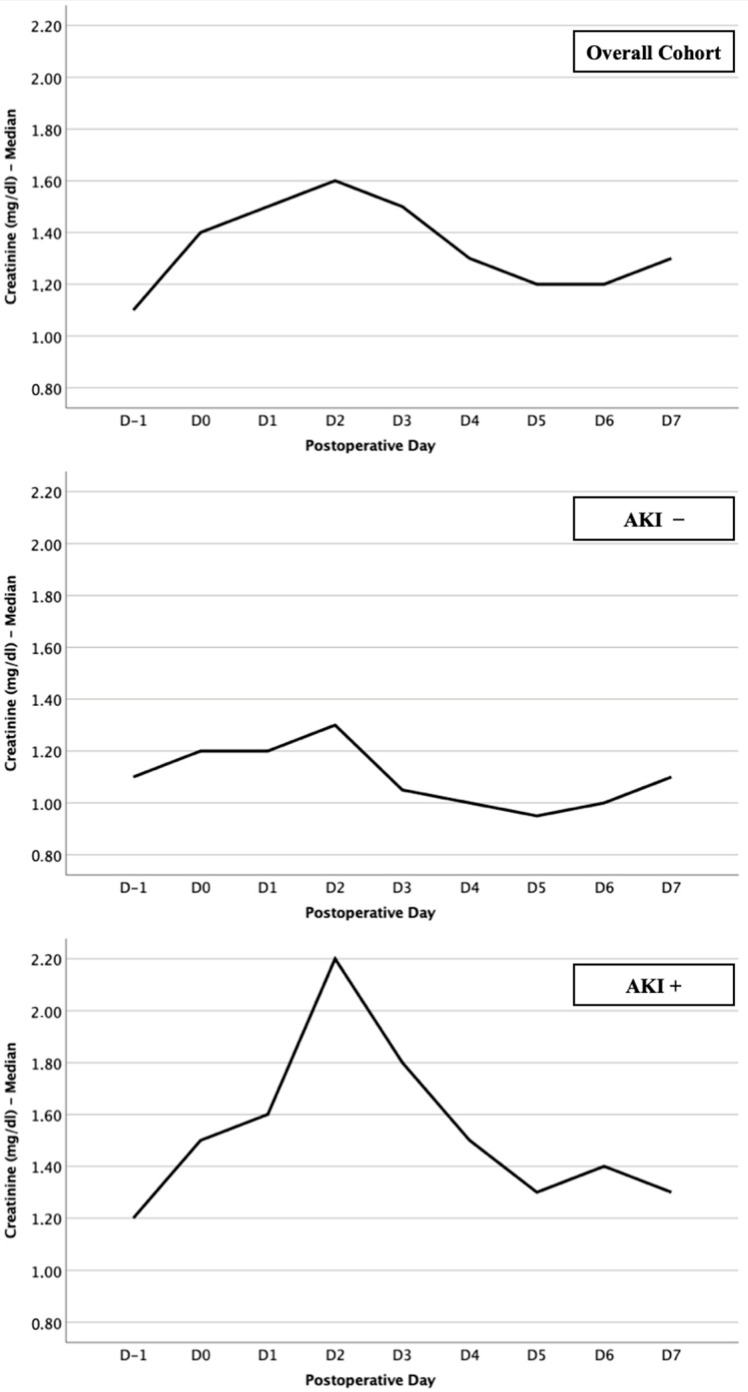
Creatinine levels in the pre- and postoperative courses after IE-related valve surgery; 1A: overall cohort; 1B: AK− patients; 1C: AK+ patients.

**Figure 2 jcm-13-04450-f002:**
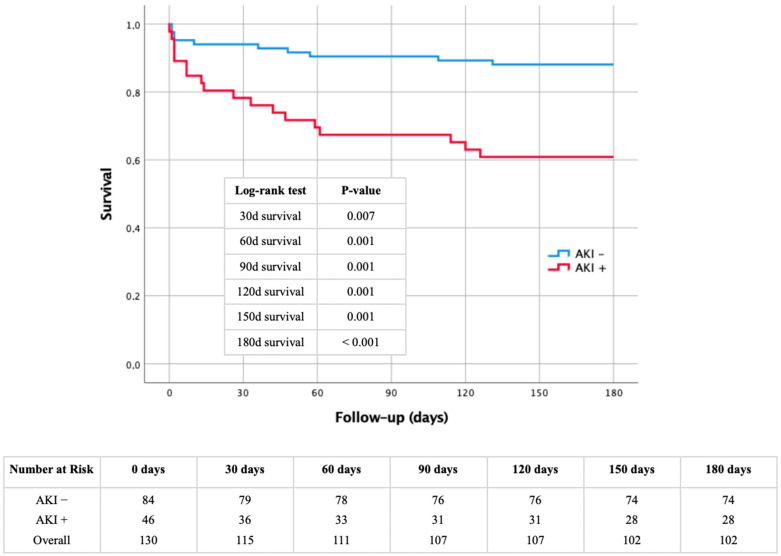
Kaplan–Meier curve with corresponding numbers at risk and log-rank tests for the detection of 30- to 180-day mortality dependent on the presence or absence regarding AKI.

**Figure 3 jcm-13-04450-f003:**
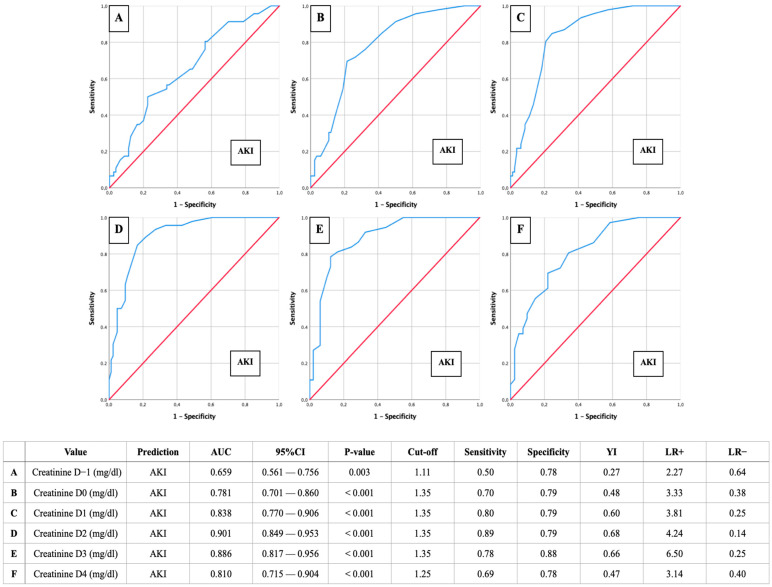
AUROC analyses of perioperative (D-1 to D4) creatinine values for the prediction of AKI with concerning cut-off values, Youden Index (YI), sensitivity, specificity, and positive and negative likelihood ratios (LR+ and LR−).

**Figure 4 jcm-13-04450-f004:**
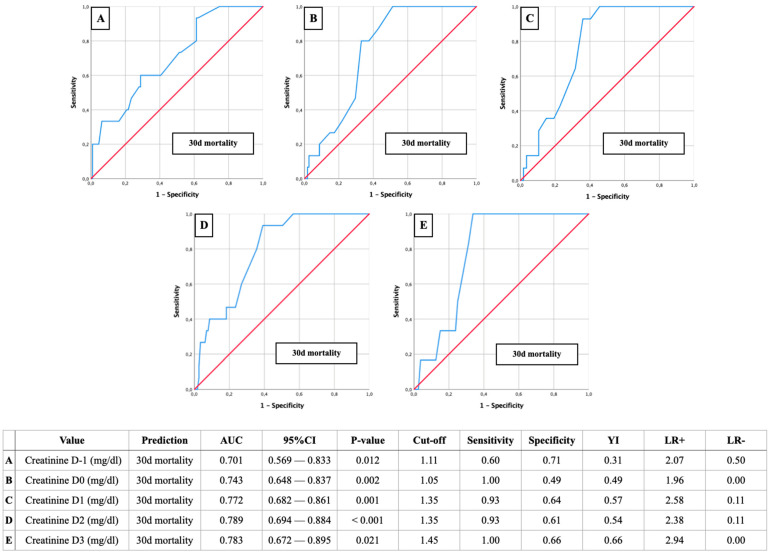
AUROC analyses of perioperative (D-1 to D3) creatinine values for the prediction of 30-day mortality with concerning cut-off values, Youden Index (YI), sensitivity, specificity, and positive and negative likelihood ratios (LR+ and LR−).

**Figure 5 jcm-13-04450-f005:**
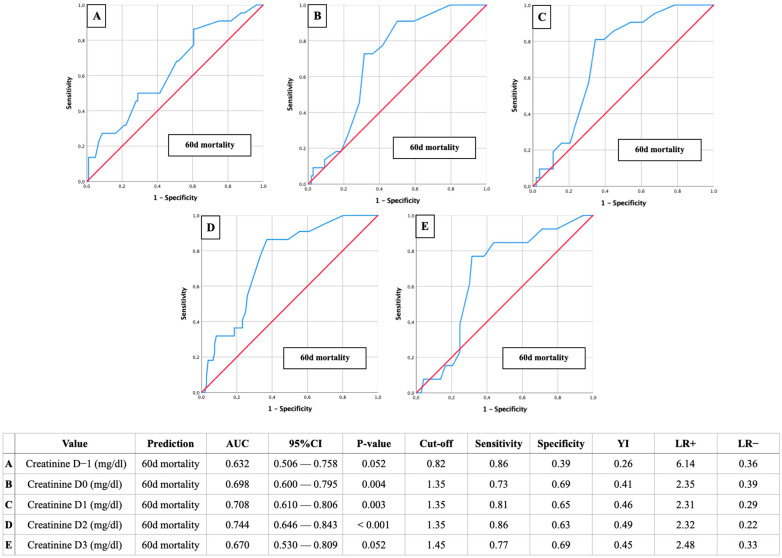
AUROC analyses of perioperative (D-1 to D3) creatinine values for the prediction of 60-day mortality with concerning cut-off values, Youden Index (YI), sensitivity, specificity, and positive and negative likelihood ratios (LR+ and LR−).

**Figure 6 jcm-13-04450-f006:**
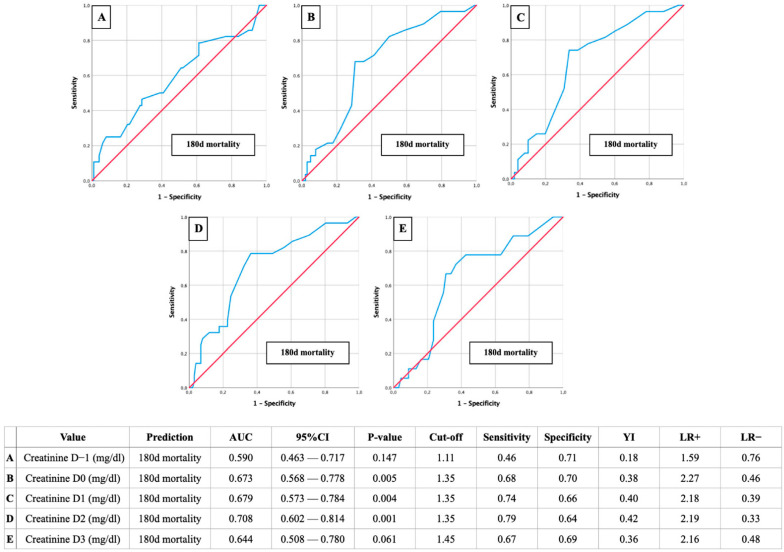
AUROC analyses of perioperative (D-1 to D3) creatinine values for the prediction of 180-day mortality with concerning cut-off values, Youden Index (YI), sensitivity, specificity, and positive and negative likelihood ratios (LR+ and LR−).

**Table 1 jcm-13-04450-t001:** Baseline characteristics of the overall study cohort and presence or absence of AKI.

	Total	AKI+	AKI−	*p*-Value
No. (%)				
Total	130 (100)	46 (35.4)	84 (64.6)	
Gender (male)	92 (70.8)	31 (67.4)	61 (72.6)	0.531
Age				
<20	1 (0.8)	0 (0.0)	1 (1.2)	0.458
20–39	14 (10.8)	3 (6.5)	11 (13.1)	0.248
40–59	29 (22.3)	6 (13.0)	23 (27.4)	0.060
60–79	81 (62.3)	35 (76.1)	46 (54.8)	0.016
≥80	5 (3.8)	2 (4.3)	3 (3.6)	0.826
BMI				
<18.5	2 (1.5)	0 (0.0)	2 (2.4)	0.292
18.5–24.9	50 (38.5)	14 (30.4)	36 (42.9)	0.164
25.0–29.9	55 (42.3)	21 (45.7)	34 (40.5)	0.568
30.0–34.9	17 (13.1)	7 (15.2)	10 (11.9)	0.592
35.0–39.9	6 (4.6)	4 (8.7)	2 (2.4)	0.101
≥40.0	0 (0.0)	0 (0.0)	0 (0.0)	-
NYHA				
NYHA I	78 (60.0)	25 (54.3)	53 (63.1)	0.330
NYHA II	29 (22.3)	8 (17.4)	21 (25.0)	0.319
NYHA III	12 (9.2)	8 (17.4)	4 (4.8)	0.017
NYHA IV	11 (8.5)	5 (10.9)	6 (7.1)	0.465
Microbiology				
*Staphylococcus* spp.	48 (36.9)	22 (47.8)	26 (31.0)	0.057
*Staphylococcus aureus*	37 (28.5)	18 (39.1)	19 (22.6)	0.046
*Staphylococcus epidermidis*	6 (4.6)	2 (4.3)	4 (4.8)	0.914
*Streptococcus* spp.	29 (22.3)	4 (8.7)	25 (29.8)	0.006
*Streptococcus mitis/oralis*	10 (7.7)	2 (4.3)	8 (9.5)	0.290
*Streptococcus sanguis/parasanguis*	9 (6.9)	1 (2.2)	8 (9.5)	0.114
*Enterococcus* spp.	17 (13.1)	7 (15.2)	10 (11.9)	0.592
*Enterococcus faecalis*	16 (12.3)	7 (15.2)	9 (10.7)	0.455
HACEK group	1 (0.8)	0 (0.0)	1 (1.2)	0.458
*Candida* spp.	1 (0.8)	0 (0.0)	1 (1.2)	0.458
Polymicrobial IE	8 (6.2)	4 (8.7)	4 (4.8)	0.372
Others	5 (3.8)	0 (0.0)	5 (6.0)	0.458
Negative Blood Cultures and PCRs	21 (16.2)	9 (19.6)	12 (14.3)	0.434
Pre-existing Conditions				
Diabetes mellitus	19 (14.6)	5 (10.9)	14 (16.7)	0.371
Arterial Hypertension	66 (50.8)	26 (56.5)	40 (47.6)	0.332
CVD	40 (30.8)	14 (30.4)	26 (31.0)	0.951
Previous Myocardial Infarction	8 (6.2)	3 (6.5)	5 (6.0)	0.897
Atrial fibrillation	35 (26.9)	16 (34.8)	19 (22.6)	0.135
Previous Aortocoronary Bypass	15 (11.5)	7 (15.2)	8 (9.5)	0.331
Pacemaker (before IE)	9 (6.9)	3 (6.5)	6 (7.1)	0.894
COPD	8 (6.2)	6 (13.0)	2 (2.4)	0.016
Nicotine Consumption	15 (11.5)	4 (8.7)	11 (13.1)	0.453
Hyperlipidemia	53 (40.8)	22 (47.8)	31 (36.9)	0.226
Stroke (before Endocarditis)	11 (8.5)	5 (10.9)	6 (7.1)	0.465
PAD	8 (6.2)	3 (6.5)	5 (6.0)	0.897
Chronic Kidney Disease	18 (13.8)	6 (13.0)	12 (14.3)	0.845
Chronic Heart Failure	22 (16.9)	5 (10.9)	17 (20.2)	0.173
Premedication				
Beta-Blocker	58 (44.6)	23 (50.0)	35 (41.7)	0.361
Diuretics	56 (43.1)	25 (54.3)	31 (36.9)	0.055
ACEI/ARB/ARNI	36 (27.7)	17 (37.0)	19 (22.6)	0.081
Statins	33 (25.4)	16 (34.8)	17 (20.2)	0.068
Preoperative Echocardiography				
Aortic Valve Stenosis I°	7 (5.4)	3 (6.5)	4 (4.8)	0.671
Aortic Valve Stenosis II°	6 (4.6)	4 (8.7)	2 (2.4)	0.101
Aortic Valve Stenosis III°	5 (3.8)	1 (2.2)	4 (4.8)	0.463
Aortic Valve Insufficiency I°	27 (20.8)	9 (19.5)	18 (21.4)	0.802
Aortic Valve Insufficiency II°	21 (16.2)	8 (17.4)	13 (15.5)	0.777
Aortic Valve Insufficiency III°	29 (22.3)	8 (17.4)	21 (25.0)	0.319
Mitral Valve Stenosis I°	4 (3.1)	2 (4.3)	2 (4.3)	0.826
Mitral Valve Stenosis II°	1 (0.8)	1 (2.2)	0 (0.0)	0.638
Mitral Valve Stenosis III°	0 (0.0)	0 (0.0)	0 (0.0)	-
Mitral Valve Insufficiency I°	55 (42.3)	19 (41.3)	36 (42.9)	0.826
Mitral Valve Insufficiency II°	30 (23.1)	14 (30.4)	16 (19.0)	0.864
Mitral Valve Insufficiency III°	32 (24.6)	9 (19.6)	23 (27.4)	0.323
Pulmonary Valve Stenosis	0 (0.0)	0 (0.0)	0 (0.0)	-
Pulmonary Valve Insufficiency I°	26 (20.0)	11 (23.9)	15 (17.8)	0.409
Pulmonary Valve Insufficiency II°	0 (0.0)	0 (0.0)	0 (0.0)	-
Pulmonary Valve Insufficiency III°	2 (1.5)	1 (2.2)	1 (1.2)	0.663
Tricuspid Valve Stenosis	0 (0.0)	0 (0.0)	0 (0.0)	-
Tricuspid Valve Insufficiency I°	64 (49.2)	15 (50.0)	41 (48.8)	0.897
Tricuspid Valve Insufficiency II°	12 (9.2)	5 (10.9)	7 (8.3)	0.633
Tricuspid Valve Insufficiency III°	6 (4.6)	3 (6.5)	3 (3.6)	0.443
Preoperative Antimicrobial Therapy				
Beta-Lactam (Penicillins)	69 (53.1)	31 (67.4)	38 (45.2)	0.016
Beta-Lactam (Cephalosporins)	59 (45.4)	20 (43.5)	39 (46.4)	0.747
Beta-Lactam (Carbapenems)	2 (1.5)	1 (2.2)	1 (1.2)	0.663
Ansamycine	21 (16.2)	12 (26.1)	9 (10.7)	0.023
Glycopeptide	15 (11.5)	5 (10.9)	10 (11.9)	0.860
Aminoglycoside	10 (7.7)	3 (6.5)	7 (8.3)	0.711
Lipopeptide	5 (3.8)	1 (2.2)	4 (4.8)	0.463
Lincosamide	3 (2.3)	1 (2.2)	2 (2.4)	0.940
Phosphonic Antibiotics	2 (1.5)	0 (0.0)	2 (2.4)	0.292
Oxazolidinone	1 (0.8)	1 (2.2)	0 (0.0)	0.175
Tetracycline	1 (0.8)	0 (0.0)	1 (1.2)	0.458
Preoperative Conditions				
Elective Surgery	9 (6.9)	2 (4.3)	7 (8.3)	0.392
Urgent Surgery	97 (74.6)	35 (76.1)	62 (73.8)	0.775
Emergency Surgery	24 (18.5)	9 (19.6)	15 (17.9)	0.810
Prosthetic Valve Endocarditis	35 (26.9)	16 (34.8)	19 (22.6)	0.135
Cardiogenic shock	1 (0.8)	1 (2.2)	0 (0.0)	0.175
Intraoperative Conditions				
Endocarditis of One Heart Valve	111 (85.4)	38 (82.6)	73 (86.9)	0.507
Endocarditis of Two Heart Valves	19 (14.6)	8 (17.4)	11 (13.1)	0.525
Endocarditis of Three Heart Valves	0 (0.0)	0 (0.0)	0 (0.0)	-
One Surgically Repaired Heart Valve	79 (60.8)	23 (50.0)	54 (66.7)	0.063
Two Surgical Repaired Heart Valves	43 (33.1)	18 (39.1)	25 (29.8)	0.278
Three Surgical Repaired Heart Valves	8 (6.2)	5 (10.9)	3 (3.6)	0.098
Additional aortocoronary bypass	11 (8.5)	5 (10.9)	6 (7.1)	0.465
Cardioplegia	124 (95.4)	44 (95.7)	80 (95.2)	0.914
Blood Products	94 (72.3)	40 (87.0)	54 (64.3)	0.006
Postoperative Conditions				
ECMO	5 (3.8)	3 (6.5)	2 (2.4)	0.240
Bleeding/Tamponade	15 (11.5)	6 (13.0)	9 (10.7)	0.691
Stroke	4 (3.1)	2 (4.3)	2 (2.4)	0.535
Valvular Complications	1 (0.8)	1 (0.8)	0 (0.0)	0.175
Complicated Pneumonia	5 (3.8)	4 (8.7)	1 (1.2)	0.033
Wound Healing Disorder	6 (4.6)	4 (8.7)	2 (2.4)	0.101
Third-Degree Atrioventricular Block	15 (11.5)	4 (8.7)	11 (13.1)	0.453
Sepsis	2 (1.5)	0 (0.0)	2 (2.4)	0.292
Tracheostomy	4 (3.1)	2 (4.3)	2 (2.4)	0.535
In-Hospital Death	21 (16.2)	15 (32.6)	6 (7.1)	<0.001
Mean ± SD				
Age (years)	61.9 ± 14.4	65.7 ± 12.2	59.8 ± 15.1	0.023
Height (cm)	172.9 ± 7.8	171.9 ± 7.4	173.5 ± 8.0	0.265
Weight (kg)	79.4 ± 15.6	82.2 ± 15.5	77.9 ± 15.5	0.135
BMI (kg/m^2^)	26.5 ± 4.5	27.8 ± 4.6	25.8 ± 4.3	0.016
BSA (m^2^)	1.9 ± 0.2	1.9 ± 0.2	1.9 ± 0.2	0.403
ACEF 2 Score	3.1 ± 1.6	3.3 ± 1.5	3.0 ± 1.6	0.300
EuroScore II	10.4 ± 10.2	13.4 ± 9.6	8.8 ± 10.3	0.013
Days from Admission to Surgery (d)	4.0 ± 5.8	3.7 ± 5.9	4.2 ± 5.8	0.633
Surgery Time (min)	271.8 ± 114.8	316.4 ± 117.0	247.3 ± 106.6	0.001
Clamping Time (min)	104.0 ± 53.9	125.6 ± 58.9	92.1 ± 47.3	0.001
Perfusion Time (min)	158.4 ± 86.2	194.0 ± 96.5	138.9 ± 73.6	<0.001
Hospitalization Days (d)	20.6 ± 20.3	23.7 ± 22.6	19.0 ± 18.9	0.208
Postoperative Days (d)	24.5 ± 21.7	27.2 ± 24.8	23.1 ± 19.9	0.302
Ventilation Period (h)	55.5 ± 87.7	91.7 ± 99.9	35.7 ± 73.6	0.001
ICU stay (h)	240.8 ± 390.5	353.4 ± 493.8	179.1 ± 306.5	0.033
Red Blood Cell Concentrates (No.)	1.9 ± 2.7	2.2 ± 1.8	1.7 ± 3.1	0.402
Platelet Concentrate (No.)	0.4 ± 0.8	0.6 ± 0.9	0.3 ± 0.8	0.060
FFPs (No.)	1.1 ± 2.1	1.9 ± 2.5	0.7 ± 1.7	0.004
Median ± IQR				
LVEF (%)	55.0 ± 5.0	55.0 ± 4.5	55.0 ± 5.0	0.746
Min. Hb—intraop. (g/dL)	7.5 ± 1.0	7.4 ± 1.1	7.6 ± 1.3	0.056
Min. Hb—6 h postop. (g/dL)	9.1 ± 1.7	8.4 ± 1.3	9.5 ± 1.7	0.077
Min. Hb—24 h postop. (g/dL)	8.7 ± 1.5	7.9 ± 1.1	8.9 ± 1.2	0.075
Max. Lactate—6 h postop. (mmol/L)	2.6 ± 2.5	3.8 ± 3.3	2.3 ± 1.4	0.006
Max. Lactate—24 h postop. (mmol/L)	2.9 ± 2.8	4.6 ± 2.1	2.4 ± 2.2	0.023
Max. Troponin T—24 h postop. (ng/L)	800.0 ± 1380.5	1024.0 ± 3619.5	728.5 ± 798.3	0.007
Max. CK-MB—24h postop (U/L)	57.8 ± 54.8	91.7 ± 117.1	53.6 ± 28.3	<0.001
Min. MAP—intraop. (mmHg)	47.0 ± 7.1	47.0 ± 7.1	49.0 ± 7.5	0.439
Max. NOR—intraop. (mL/min/kg)	0.3 ± 0.2	0.3 ± 0.2	0.3 ± 0.2	0.149
⌀ NOR—intraop. (mL/min/kg)	0.2 ± 0.1	0.2 ± 0.2	0.2 ± 0.1	0.282
Renal Excretion 1–2 h (mL)	40.0 ± 65.0	30.0 ± 22.5	45.0 ± 103.8	0.001
Renal Excretion 2–3 h (mL)	40.0 ± 52.5	20.0 ± 30.0	50.0 ± 70.0	<0.001
Drainage Volume—6 h postop. (mL)	230.0 ± 285.0	350.0 ± 215.0	150.0 ± 145.0	<0.001
Drainage Volume—12 h postop. (mL)	270.0 ± 350.0	350.0 ± 287.5	150.0 ± 312.5	<0.001
Drainage Volume—24 h postop. (mL)	550 ± 587.5	800.0 ± 512.5	400 ± 548.8	0.008
Fluide Volume—intraop. (L)	3.2 ± 1.6	3.2 ± 1.6	3.2 ± 1.6	0.134

AKI: acute kidney injury; BMI: body mass index; NYHA: New York Heart Association; CVD: cardiovascular disease; IE: infective endocarditis; COPD: chronic obstructive pulmonary disease; PAD: peripheral arterial occlusive disease; ACEI: angiotensin-converting enzyme inhibitor; ARB: angiotensin receptor blocker; ARNI: angiotensin receptor-neprilysin inhibitor; ECMO: extracorporeal membrane oxygenation; BSA: body surface area; ICU: intensive care unit; FFP: fresh frozen plasma; LVEF: left ventricular ejection fraction; Min.: minimal; Hb: hemoglobin; Max.: maximum; CK-MB: creatine phosphokinase-MB; MAP: mean arterial pressure; NOR: noradrenaline. ⌀: average.

**Table 2 jcm-13-04450-t002:** Overview of the perioperative creatinine courses (D-1 to D7) in the entire cohort and depending on the presence of AKI.

	Total	AKI−	AKI+	*p*-Value
Median ± IQR				
Creatinine D-1 (mg/dL)	1.1 ± 0.7	1.1 ± 0.6	1.2 ± 0.7	0.003
Creatinine D0 (mg/dL)	1.4 ± 0.8	1.2 ± 0.7	1.5 ± 0.9	<0.001
Creatinine D1 (mg/dL)	1.5 ± 0.8	1.2 ± 0.6	1.6 ± 0.9	<0.001
Creatinine D2 (mg/dL)	1.6 ± 1.0	1.3 ± 0.7	2.2 ± 1.0	<0.001
Creatinine D3 (mg/dL)	1.5 ± 1.0	1.1 ± 0.7	1.8 ± 1.0	<0.001
Creatinine D4 (mg/dL)	1.3 ± 0.9	1.0 ± 0.7	1.5 ± 1.0	<0.001
Creatinine D5 (mg/dL)	1.2 ± 0.7	1.0 ± 0.7	1.3 ± 0.9	<0.001
Creatinine D6 (mg/dL)	1.2 ± 1.0	1.0 ± 0.7	1.4 ± 1.2	<0.001
Creatinine D7 (mg/dL)	1.3 ± 0.8	1.1 ± 0.7	1.3 ± 1.6	0.003

**Table 3 jcm-13-04450-t003:** Univariate and multivariable binary logistic regression analysis detecting AKI in patients with IE undergoing valve surgery.

AKI Binary Logistic Regression	Univariate		Multivariable	
	Hazard Ratio (95% CI)	*p*-Value	Hazard Ratio (95% CI)	*p*-Value
Age	1.601 (1.057–2.423)	0.026	0.656 (0.062–6.982)	0.727
BMI	1.572 (1.079–2.291)	0.019	2.035 (0.579–7.149)	0.268
EuroScore II	1.564 (1.080–2.265)	0.018	1.329 (0.636–2.775)	0.449
Renal Excretion 0–1 h (postoperative)	0.331 (0.117–0.942)	0.038	0.445 (0.073–2.703)	0.379
Renal Excretion 2–3 h (postoperative)	0.324 (0.141–0.744)	0.008	0.003 (0.000–0.275)	0.012
Hb minimal (intraoperative)	0.437 (0.212–0.900)	0.025	0.203 (0.044–0.926)	0.039
Surgery Time	1.873 (1.268–2.766)	0.002	0.225 (0.007–7.564)	0.405
Clamping Time	1.964 (1.292–2.985)	0.002	0.220 (0.019–2.499)	0.222
Perfusion Time	2.003 (1.316–3.048)	0.001	1.260 (0.447–3.552)	0.662
Ventilation Time	1.954 (1.285–2.972)	0.002	3.096 (0.118–81.191)	0.498
Intensive Care Unit Time	1.660 (1.036–2.660)	0.035	0.421 (0.023–7.817)	0.561
Blood Products (intraoperative)	3.704 (1.408–9.743)	0.008	0.094 (0.006–1.524)	0.096
FFP (intraoperative)	1.795 (1.232–2.616)	0.002	2.388 (0.945–6.030)	0.066
Lactate maximum (6 h postoperative)	1.607 (1.058–2.441)	0.026	00.461 (0.158–1.344)	0.156
Troponin T maximum (24 h postoperative)	2.722 (1.381–5.365)	0.004	1.193 (0.050–28.343)	0.913
CK-MB maximum (24 h postoperative)	5.483 (1.965–15.300)	0.001	10.671 (1.733–65.723)	0.011
Quantity of Surgically Treated Heart Valves	1.885 (1.045–3.400)	0.035	0.629 (0.045–8.755)	0.730

AKI: acute kidney injury; CI: confidence interval; BMI: body mass index; Hb: hemoglobin; FFP: fresh frozen plasma; CK-MB: creatine phosphokinase-MB.

## Data Availability

The data underlying this article will be shared upon reasonable request to the corresponding author.

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
