# Peer review of "Beyond the Valve: Incidence, Outcomes, and Modifiable Factors of Acute Kidney Injury in Patients with Infective Endocarditis Undergoing Valve Surgery—A Retrospective, Single-Center Study"

_jcm, 2024, doi:10.3390/jcm13154450_

Round 1
Reviewer 1 Report
Comments and Suggestions for Authors
Beyond the Valve: Incidence, Outcomes, and Modifiable Factors of Acute Kidney Injury in Patients with Endocarditis Undergoing Valve Surgery — A Retrospective, Single-Center Study PEER REVIEW This is a single-center retrospective study assessing incidence and risk factors of acute kidney injury (AKI) in patients who underwent open heart valve surgery for endocarditis. The topic is of extreme interest because kidney function can be profoundly altered during infective endocarditis and undoubtedly it may have a dramatic impact on the outcome. The sample size is small. English language mostly fine but it requires a general revision together with a check of terminology (for example: authors should refer to endocarditis as infective endocarditis in all the manuscript with the abbreviation “IE”). The abstract clearly summarizes the main findings (please change “infectious endocarditis” with infective endocarditis and be consistent in all the manuscript). Ethical approval is showed together with information regarding informed consent. I do have some major concerns.
Major revision:
1. Methods, study population: AKI before cardiac surgery was considered as an exclusion criteria? This is not clear to me and it should be stated by the authors. Many factors such as nephrotoxic antimicrobials, heart failure, renal emboli or immunological phenomena (IE related glomerulonephritis) could cause AKI before cardiac surgery in patients with IE. If authors want to focus on the incidence of post-operative AKI, patients with AKI before surgery could be considered for exclusion or analysed in a separate cohort.
2. Methods, endocarditis: the study period is 2013-2021. I believe patients were diagnosed with IE according to modified Duke criteria before 2015 and according to ESC 2015 criteria later on. Please clarify it and change references accordingly. Ref number 13 and 14 refers to ESC 2015 and ESC 2023 guidelines. Please clarify if only patients with definite IE were included. If also patients with possible IE according to existing criteria overtime were included this should be stated.
3. Methods, outcomes: I suggest differentiating primary outcome (incidence and determinants of AKI) from secondary outcome (AKI impact on mortality) 4. Results, general characteristics: please provide indications for cardiac surgery according to ESC guidelines (heart failure, uncontrolled infection or high risk of embolism). It would be of extreme interest to compare incidence of AKI in these three subgroups to understand possible drivers of kidney function in the peri-operative period and high-risk subgroups. If available, please provide the median time (in days) between admission and cardiac surgery in AKI+ vs AKI- patients.
5. Results, general characteristics: if available, authors should provide echocardiographic data such as LVEF, vegetation size, IE-related valve disease (stenosis and regurgitation with associated degrees). If these data are not available this should be mentioned in the limitation section.
6. Results general characteristics: please provide incidence of IE-related complications such as embolic events. It would be interesting to know the incidence of renal embolic events and the impact of this complication on postoperative kidney function.
7. Results, 2nd day creatinine levels: this finding is extremely interesting. If available, please describe how many patients received infusion of iodinated contrast medium before cardiac surgery and when it occurred. We all know that patients undergoing cardiac surgery for IE receive nephrotoxic contrast medium before cardiac surgery for coronary angiography or contrast-enhanced CT scan and that contrast-induced nephropathy (CIN) usually occurs after 48-72 hours of the contrast medium infusion.
Minor concerns:
1. If available, consider to include the type of antimicrobials administered before cardiac surgery to assess the possible impact of nephrotoxic medications in the occurrence of AKI.
2. TABLE 1: change “mixed infection” with “polymicrobial IE”. Change PAOD with PAD. To enhance visibility of significant p-values consider to use bold for them.
3. Figure 3 and Figure 4: please include positive and negative likelihood ratio for the several cutoff analysed.
4. It is necessary to increase quality and definition of the figures.
Comments on the Quality of English Language
Please see attached file
Reviewer 2 Report
Comments and Suggestions for Authors
Thank you for reviewing this manuscript. The authors investigated the incidence and outcomes of AKI in Patients with endocarditis. The study is interesting, although this is a vulnerable group of patients in whom the development of AKI is obvious. The study is retrospective and, therefore, leaves several questions open.
First, the etiology of AKI is unknown, and no such examination has been conducted. Determining the simplest fractional Na excretion would have helped to decide whether AKI occurred in connection with hypotension-hypovolaemia or acute tubular necrosis.
It is also unclear whether the increase in CK-MB leads to ATN rather than a marker predicting AKI. Rather, it is the significant increase in CK-MB that potentiates the process of AKI.
The results also show that a longer perfusion time significantly increases the risk of AKI.
The manuscript is relatively well written but has shortcomings that must be corrected. For example, the resolutions of abbreviations in the tables are incomplete; this must be replaced.
The quality of Figures 3 and 4 is poor, unreadable, and needs to be improved.
Round 2
Reviewer 1 Report
Comments and Suggestions for Authors
No further comments
Reviewer 2 Report
Comments and Suggestions for Authors
The authors made the corrections and answered the raised questions. The manuscript improved. I have no questions more.